# Regional political climate's moderating role in the association between political conservatism and COVID-19 vaccine hesitancy in the United States

Rachel E. Dinero[1,2], William B. Monti[1], Brittany L. Kmush[3]*

1 Department of Psychological and Brain Sciences, Colgate University, Hamilton, New York, United States of America, 2 Department of Psychology, Le Moyne College, Syracuse, New York, United States of America, 3 Department of Public Health, Syracuse University, Syracuse, New York, United States of America

* blkmush@syr.edu

## Abstract

There is an emerging body of evidence linking political conservatism and conservative political climate in the United States to COVID-19 vaccine hesitancy and uptake. The goal of the present research was to examine how political climate moderates the relationship between self-reported political conservatism and COVID-19 vaccine hesitancy and uptake. We collected online survey data from 683 participants between March 8 and April 19, 2023. Controlling for age, education, income, and race, there was an interaction between political conservatism and conservative political climate for both vaccine and booster hesitancy ($\beta = .07$, $p = .03$; $\beta = .12$, $p < .001$, respectively), such that liberals were less likely to be hesitant regardless of political climate. However, conservatives living in liberal political climates were less vaccine hesitant than their conservative counterparts living in conservative regions. A similar interaction was for the likelihood of receiving a COVID-19 booster (OR $= .84$, $p = .049$). Liberals were more likely to receive a booster regardless of political climate, while conservatives' likelihood was associated with their political climate. Observed patterns linking liberal political climates with vaccine uptake among conservative individuals have important implications for vaccination efforts among conservative individuals in the United States.

## Introduction

The COVID-19 pandemic has been unprecedented in recent history. From January 2020 until April 30, 2023, there were over 1.3 million deaths from COVID in the United States alone [1]. During that time, millions more Americans of all ages were hospitalized due to COVID with rates ranging from 179.8 per 100,00 during 2020 up to 521.2 per 100,000 during the 2021−2022 seasons [2]. Due to the high morbidity and mortality of this disease, many governments made vaccine development a

**Data availability statement:** The materials and data presented in this study are openly available in OSF at https://osf.io/s2knw/?view_only=4de7d7296d0a44acab11174e357776b.

**Funding:** This research was funded by a Public Affairs and Policy Research Initiative Grant from Colgate University. The funders had no role in study design, data collection and analysis, decision to publish, or preparation of the manuscript.

**Competing interests:** The authors have declared that no competing interests exist.

priority. The vaccines greatly reduced the morbidity and mortality caused by COVID-19, with an estimated 2.5 million deaths averted and saved 15 million life-years worldwide by 2024 [3]. However, the rapid development and rollout of the vaccines was not without controversy. The relatively small sample sizes in the trails (over 20,000 people) could not rule out the possibility of missing rare adverse events, such as myocarditis in young, healthy males. Additionally, rigorous trials have not been completed to determine additional protection provided by subsequent or yearly vaccine doses in previously exposed populations [4].

The controversy around vaccines spread from the scientific and regulatory community to politics and the public, possibly exacerbated by the timing of the election cycle in the US [5]. In the US, political conservatism has been consistently associated with greater COVID-19 vaccine hesitancy [6–12] and lower COVID-19 vaccine uptake [10,13,14]. While political conservatism has been historically associated with lower trust in the scientific community, which is subsequently linked to vaccine hesitancy [15,16], the increased partisan divide in vaccine hesitancy may be attributed, at least in part, to the politicization of COVID-19 [17]. The influence of political orientation on vaccine hesitancy has been increasing since the onset of the pandemic and political orientation has become a better predictor of vaccine hesitancy than demographic variables, pandemic-related fear, and trust in health institutions [18].

The influence of political orientation on vaccine hesitancy and uptake may extend beyond individual orientation. There is an emerging body of evidence that the broader political climate within which an individual lives can influence COVID-19 vaccine hesitancy and uptake. For example, at both the county and the state level there was a positive association between Democratic vote share (i.e., % of people voting for the Democratic candidate) in the 2020 presidential election and COVID-19 vaccination rates from 2021−2022 [19]. Likewise, state-level 2020 Democratic presidential voting share was associated with higher state-level COVID-19 booster rates in 2022 [20]. Similarly, those who lived in states with higher Republican vote shares in the 2020 presidential election were more likely to report vaccine hesitancy in a 2021 sample of 443,680 participants [21]. While previous research has identified the unique impact of both political orientation and political climate on vaccination attitudes, there has been little research on the interaction between personal political orientation and local political climate.

The present research examines political climate as a potential moderator of the association between political orientation and COVID-19 vaccine hesitancy and uptake. Here, we examined the moderating role of political climate on the association between individual political orientation and COVID-19 vaccine hesitancy.

## Materials and methods

### Study design

This was an online self-reported study in which a nonprobability convenience sample was recruited from the United States between March 8 and April 19, 2023. The online survey assessed demographic control variables (i.e., age, gender, race, income, education, perceived vaccine access), predictor variables (i.e., political conservatism,

regional political climate) and outcome variables (i.e., vaccine status, booster status, vaccine hesitancy, booster hesitancy) (S1 File).

## Predictor variables

Demographic control variables (i.e., gender, race, income, and education) were assessed using single-item multi-option questions. Perceived vaccine access was assessed using a single-item question "If I needed to get a COVID-19 vaccine or booster, I could easily get it", rated on a 5-point Likert scale from Disagree Strongly (1) to Agree Strongly (5). Political conservatism (i.e., a political ideology that emphasizes tradition, authority, and limited government intervention) was assessed across two items. The first item, assessing social conservatism (i.e., a political ideology that focuses on preserving traditional social institutions and opposing social change), asked participants to indicate their political orientation on "social issues (for example, abortion, gun rights, gay rights)". The second item, assessing economic conservatism (i.e., a political ideology that emphasizes free markets and lower taxes), aske participants to indicate their political orientation on "economic issues (for example, taxation, government spending)". Both items were rated on a 11-point sliding scale from strongly liberal (0) to strongly conservative (10). These two items were averaged to form a political conservatism scale.

While conservative ideology refers to individual beliefs, conservative regional climate refers to the broader ideological and partisan context in which an individual resides. We assessed regional political climate using zip code data provided by participants and 2020 presidential election results sourced from the New York Times [22]. Zip codes are postal codes used by the U.S. postal service to identify geographic delivery areas. These regions are often used as a proxy for local community characteristics [23]. Election results included the numbers of votes for the Democratic and Republican candidate (Biden and Trump, respectively) as well as the total number of votes for each Federal Information Processing Standard (FIPS) code. Political climate for each participant was determined by matching FIPS codes to corresponding zip codes. In cases where a zip code included multiple FIPS codes, data was averaged across these FIPS. Using the FIPS data for each zip code, we calculated the percent of Republican votes by dividing the number of Republican votes by the total number of votes (multiplied by 100). This reflects measurement processes in prior research [10,19,21]. We used this % Republican vote by zip code as a measure of conservative regional climate. Participants who did not provide zip code data were removed from the dataset.

## Outcome variables

Self-reported vaccine status and booster status were assessed through individual items, "have you received a COVID-19 vaccine" and "have you received a COVID-19 booster". The term "booster" is used in this analysis as this was the common term at the time to refer to subsequent doses of the COVID-19 vaccine after the initial vaccine series. Possible responses were yes, no, and I don't know. Vaccine hesitancy (i.e., negative attitudes toward vaccination) was measured using the seven items from the Attitudes towards Adult Vaccination Scale [24], which assesses the perceived value of vaccines in general. We modified the items to assess attitudes specifically toward the COVID-19 vaccine [e.g., I fear the potential impact of the COVID-19 vaccine on my health in the future, I believe that the benefits of COVID-19 vaccination outweigh the potential risks (reverse-scored)]. Each item was rated on a 5-point Likert scale ranging from disagree strongly (1) to agree strongly (5). These items were averaged to form a vaccine hesitancy scale. Booster hesitancy was assessed using items from a 2022 survey on COVID-19 booster hesitancy [25].

## Participant recruitment

Participants were a convenience sample recruited through Qualtrics Panels, an online research platform that manages participant recruitment and data quality for academic studies. Qualtrics Panels recruits participants through a network of verified online panel providers. Individuals voluntarily join these panels and provide demographic and behavioral information, which allows Qualtrics Panels to match participants to studies that fit specific eligibility criteria. Qualtrics Panels

are widely used in academic research and have been shown empirically to produce samples with comparable demographic and behavioral characteristics to community and other panel sources when appropriate attention checks are used [26].

The data used in this study were part of a larger project assessing the impact of racial and socioeconomic marginalization on COVID-19 vaccine attitudes and behaviors [27]. The present research presents a novel analysis of the dataset from this project, focusing on political orientation and political climate, rather than marginalization. However, because of the initial goals in data collection, the sample was recruited across four quota groups based on race and income. Half of the sample was requested to be White (and no other racial identity), and half the sample was requested to be Black (and no other racial identity). Within each racial group, half of the participants were requested to be below the median annual income in the US [$70,000 [28]]. This resulted in four quota groups: White participants above-median-income, White participants below-median income, Black participants above-median income, and Black participants below-median income. Additionally, within each quota group it was requested that no more than 60% of the participants be of any gender.

No other exclusion criteria were specified; however, Qualtrics Panels does require participants to demonstrate consistent quality responding in order to remain in the panels system. Therefore, Qualtrics Panels screened out participants based on poor survey completion history. Qualtrics Panels screens data as it is collected and identifies participants that have data quality problems (e.g., no variance in responding, fast survey completion times that indicate participants are not reading items [more than two standard deviations below the mean completion time]. missing more than 25% of responses, passing required attention checks). To ensure data quality, we only include participants in our analysis that pass both Qualtrics Panels data quality screening and our own attention checks (S1 File), as recommended by previous research using Qualtrics Panels [26]. Qualtrics Panels conducted all data quality screens and marked participant's data as either passing or not passing all quality screens.

## Research ethics

Ethical approval was obtained prior to data collection from the first author's institutional review board (protocol ER-F22-35, approved on 1 October 2022) and all research was conducted in accordance with the Declaration of Helsinki. All participants completed an informed, written consent as part of the online survey.

## Statistical analyses

We performed confirmatory factor analysis on the items for the vaccine and booster hesitancy scales (S2 File) and the subsequent reliability of the scales was calculated using Cronbach's alphas. Pearson correlations were used to assess correlations between all non-nominal variables (i.e., age, income, education, perceived vaccine access, vaccine and booster hesitancy, political conservatism, and conservative regional climate). Mean differences in categorical predictors (i.e., race and gender) across booster status and vaccine status were assessed using $t$-tests, and differences in frequencies of vaccine status and booster status by gender and race were assessed using $X^2$. Independent samples t-tests were used to assess differences in all non-nominal variables across vaccinated and unvaccinated participants. The same analyses were run to assess differences between boosted and unboosted participants. Multiple linear regression models were built to assess the association between political conservatism and conservative regional climate with vaccine hesitancy, controlling for age, gender, education, income, race, and perceived vaccine access. Ordinary least squares estimation was used with default standard errors. Model 1 included only control variables age, gender, education, income, race, and vaccine access as dependent variables. Model 2 included control variables and political conservatism and conservative regional climate. Model 3 included all variables from Model 2 and the interaction between political conservatism and conservative regional climate. This interaction assessed the extent to which the association between political conservatism and vaccine attitudes/behavior was moderated by conservative regional climate. The

same process was used with booster hesitancy as the outcome variable to assess the association between political conservatism and conservative regional climate with booster hesitancy, controlling for age, gender, education, income, race, and perceived vaccine access. Additionally, we ran two sets of generalized linear models with logit link and model-based standard errors predicting vaccine status and booster status from the same three models described above. For all analyses, any participants with missing data on any of these variables were excluded from the analysis. All analysis and data visualization was conducted with R version 4.4.1 [29]. Two-sided p-values less than 0.05 were considered statistically significant.

## Results

Qualtrics Panels collected data from 1777 participants, and identified 970 participants as not passing all quality screens (described above). We eliminated an additional nine participants whose open-ended responses did not make logical sense, resulting in a sample of 798 validated participants. We eliminated an additional 115 participants who did not provide valid zip code data (N = 100) or whose zip code did not have corresponding election data (N = 15). The final sample for these analyses included 683 participants. We conducted confirmatory factor analysis on the vaccine and booster hesitancy items (S2 File). The vaccine hesitancy items demonstrated adequate reliability ($\alpha$ = .93, 95% CI [.92,.93]) and were averaged to form the vaccine hesitancy scale. The booster hesitancy items demonstrated adequate reliability ($\alpha$ = .93, 95% CI [.93,.94]) and were averaged to form the booster hesitancy scale.

### Participant characteristics

Participants ranged in age from 18 to 94 years (M = 46.29, SD = 18.94). Demographic characteristics of the sample are shown in Table 1. 519 (76%) of participants reported receiving the initial COVID-19 vaccine and 365 (53%) reported receiving at least one COVID-19 booster. Both vaccine hesitancy and booster hesitancy scores ranged from 1 to 5 out a possible 5 ($M_{vaccine}$ = 2.39, $SD_{vaccine}$ = 1.27; $M_{booster}$ = 2.47, $SD_{booster}$ = 1.32). Economic and social conservatism scores ranged from 0 to 10 out of a possible 10 ($M_{economic}$ = 5.22, SD = 3.18; $M_{social}$ = 5.00, SD = 3.22). The % Republican vote by zip code ranged from 6.53% to 88.63% (M = 45.08%, SD = 18.09%). Frequency distributions for vaccine hesitancy, booster hesitancy, political orientation, and % Republican vote are shown in Fig 1 and S3 Table.

### Individual associations between predictors and outcomes

As shown in Table 2, there was no significant differences based on gender or race for vaccine hesitancy, booster hesitancy, vaccine status, or booster status. Vaccine hesitancy was positively correlated with booster hesitancy (r = .84, p < .001), economic conservatism (r = .27, p < .001), social conservatism (r = .31, p < .001), political conservatism (r = .30, p < .001), and conservative regional climate (r = .20, p < .001); and negatively correlated with age (r = −.16, p < .001), income (r = −.17, p < .001), and education (r = −.21, p < .001). Booster hesitancy positively correlated with economic conservatism (r = .22, p < .001), social conservatism (r = .25, p < .001), political conservatism (r = .25, p < .001), and conservative regional climate (r = .20, p < .001); and negatively correlated with age (r = −.15, p < .001), income (r = −.19, p < .001), education (r = −.19, p < .001). Unvaccinated participants were significantly younger (t = −3.54, p < .001) and reported lower income (t = −4.58, p < .001) as compared to vaccinated participants. Additionally, unvaccinated participants reported higher vaccine hesitancy (t = 18.18, p < .001) and booster hesitancy (t = 16.62, p < .001), and were higher in economic (t = 3.40, p < .001), social (t = 4.58, p < .001), and overall political conservatism (t = 4.21, p < .001) and lived in more conservative regional climates (t = 4.61, p < .001). Similarly, participants who reported not receiving a booster were significantly younger (t = −6.93, p < .001) and reported lower income (t = −6.17, p < .001) as compared to boosted participants. Unboosted participants also reported higher average vaccine hesitancy (t = 16.20, p < .001), booster hesitancy (t = 2.37, p < .001), economic conservatism (t = 2.80, p = .02), social conservatism (t = 18.82, p < .001), and overall political conservatism (t = 2.72, p < .001), and lived in more conservative regional climates (t = 3.98, p < .001).

**Table 1. Participant age, gender, race, vaccine status, booster status, income, and education, United States, 2023 (N = 683).**

| Gender | N |
| --- | --- |
| Female | 345 (50.5%) |
| Male | 325 (47.6%) |
| Nonbinary/Transgender | 11 (1.6%) |
| No Response | 2 (0.3%) |
| **Race** | |
| Black | 342 (50%) |
| White | 341 (50%) |
| **Vaccine Status** | |
| Yes | 519 (75.9%) |
| No | 162 (23.7%) |
| No Response | 3 (0.4%) |
| **Booster Status** | |
| Yes | 365 (53.4%) |
| No | 308 (45.1%) |
| No Response | 10 (1.5%) |
| **Income** | |
| under $10,000 | 44 (6.4%) |
| $10,000 to $19,999 | 59 (8.6%) |
| $20,000 to $29,999 | 66 (9.7%) |
| $30,000 to $39,999 | 52 (7.6%) |
| $40,000 to $49,999 | 44 (6.4%) |
| $50,000 to $59,999 | 47 (6.9%) |
| $60,000 to $69,999 | 28 (4.1%) |
| $70,000 to $79,999 | 67 (9.8%) |
| $80,000 to $89,999 | 41 (6.0%) |
| $90,000 to $99,999 | 43 (6.3%) |
| $100,000 to $109,999 | 41 (6.0%) |
| $110,000 to $119,999 | 13 (1.9%) |
| $120,000 to $129,999 | 27 (4.0%) |
| $130,000 to $139,999 | 15 (2.2%) |
| $140,000 to $149,999 | 23 (3.4%) |
| $150,000 to $159,999 | 23 (3.4%) |
| $160,000 to $169,999 | 3 (0.4%) |
| $170,000 to $179,999 | 5 (.7%) |
| $180,000 to $189,999 | 1 (0.2%) |
| $190,000 to $199,999 | 16 (2.3%) |
| $200,000 or over | 25 (3.7%) |
| **Education** | |
| Below Primary School | 1 (0.2%) |
| Primary School | 6 (1%) |
| Secondary School | 8 (1%) |
| High School Graduate | 150 (22%) |
| Trade, Technical, or Vocational Training | 39 (6%) |
| Some College | 183 (27%) |
| Bachelor's Degree | 176 (26%) |

*(Continued)*

**Table 1.** (Continued)

| Gender | N |
|---|---|
| Master's Degree | 89 (13%) |
| Professional Degree | 10 (1%) |
| Doctoral Degree | 21 (3%) |

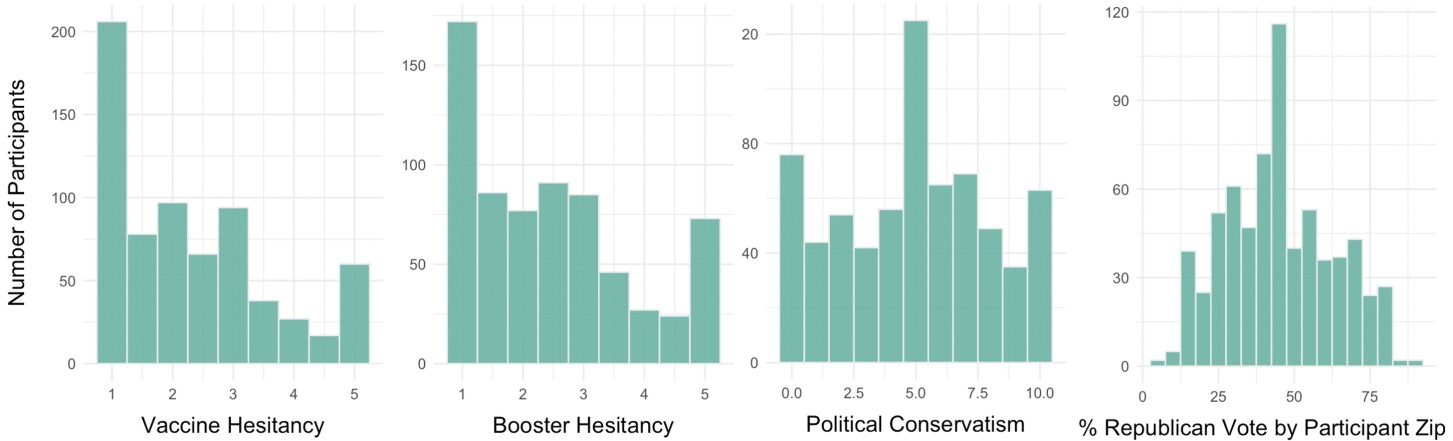

**Fig 1. Frequency distributions for vaccine hesitancy, booster hesitancy, political conservatism, and % of Republican vote lead in the 2020 presidential election based on participant zip code (N = 683).** Four histograms depicting frequency distributions for vaccine hesitancy, booster hesitancy, political conservatism, and percent Republican vote by participant zip code.

## Regression analysis

Regression analysis (Table 3) indicated that when controlling for age, education, income, race, and vaccine access, political conservatism and conservative regional climate were positively associated with vaccine hesitancy ($\beta = .23$, $p < .001$; $\beta = .11$, $p = .001$; respectively). Additionally (Table 4), the interaction between political conservatism and conservative regional climate was significant ($\beta = .09$, $p = .003$)(Table 3, Fig 2). As shown in Table 3, political conservatism and conservative regional climate were also positively associated with booster hesitancy ($\beta = .18$, $p < .001$; $\beta = .11$, $p = .002$; respectively) and the interaction between these two variables with booster hesitancy was significant ($\beta = .09$, $p = .003$)(Table 4, Fig 2). As shown in Table 5, political conservatism and conservative regional climate were negatively associated with vaccine status [odds ratio (OR) =.68, $p < .001$; OR =.72, $p = .004$; respectively]. The interaction between political conservatism and conservative regional climate (Table 6) was not significant (OR = 1.02, $p = .833$). As shown in Table 5, political conservatism and conservative regional climate were negatively associated with booster status (OR =.82, $p = .022$; OR =.77, $p = .004$; respectively). Additionally, there was a significant interaction (Table 6, Fig 2) between political conservatism and conservative regional climate (OR =.83, $p = .026$). Results for all regression models with control variables can be found in Tables A-D in S4 File.

## Discussion

The present research provides the first assessment of the potential interaction between political orientation, political climate, and COVID vaccination. Additionally, by looking at both COVID vaccine and booster status and hesitancy, we identify how attitudes and behaviors may have shifted as we transitioned from initial vaccine doses to boosters. Taken

**Table 2. Vaccine hesitancy, booster hesitancy, vaccine status and booster status associations with sex, race, age, income, education, vaccine hesitancy, booster hesitancy, economic conservatism, social, conservatism, overall political conservatism, and political climate (% Republican vote by region), United States, 2023 (N = 683).**

| | Vaccine Hesitancy | | Booster Hesitancy | | Vaccine Status | | | Booster Status | | |
|---|---|---|---|---|---|---|---|---|---|---|
| | M | t | M | t | Vaccinated (N=519) N | Not Vaccinated (N=162) N | $X^2$ | Boosted (N=365) N | Not Boosted (N=309) N | $X^2$ |
| **Sex** | | | | | | | | | | |
| Female | 2.45 | −1.04 | 2.53 | −1.37 | 263 | 81 | .00 | 174 | 164 | 2.11 |
| Male | 2.34 | | 2.39 | | 248 | 77 | | 186 | 138 | |
| **Race** | | | | | | | | | | |
| Black | 2.33 | 1.27 | 2.41 | 1.09 | 258 | 81 | .00 | 172 | 162 | 1.68 |
| White | 2.45 | | 2.52 | | 261 | 81 | | 193 | 147 | |
| | r | p | r | p | M | M | t | M | M | t |
| Age | −.16 | <.001 | −.15 | <.001 | 47.78 | 41.80 | −3.54* | 50.93 | 41.13 | −6.93* |
| Income[1] | −.17 | <.001 | −.19 | <.001 | 8.56 | 6.40 | −4.58* | 9.22 | 6.74 | −6.17* |
| Education[2] | −.21 | <.001 | −.19 | <.001 | 6.32 | 5.46 | −6.09* | 6.48 | 5.70 | −6.39* |
| Vaccine Hesitancy | – | – | .84 | <.001 | 1.99 | 3.69 | 18.18* | 1.78 | 3.13 | 16.20* |
| Booster Hesitancy | .84 | <.001 | – | – | 2.05 | 3.78 | 16.62* | 1.76 | 3.32 | 18.82* |
| Economic Conservatism | .27 | <.001 | .22 | <.001 | 5.97 | 4.99 | 3.40* | 5.54 | 4.96 | 2.37a |
| Social Conservatism | .31 | <.001 | .25 | <.001 | 6.01 | 4.67 | 4.58* | 5.39 | 4.70 | 2.80* |
| Political Conservatism | .30 | <.001 | .25 | <.001 | 4.84 | 5.99 | 4.21* | 4.83 | 5.47 | 2.72* |
| Conservative Climate | .21 | <.001 | .20 | <.001 | 43.32% | 50.73% | 4.61* | 42.58% | 48.09% | 3.98* |

*p < .001

a p = .02

[1]Income was assessed using $10,000 income brackets ranging from 1 (below $10,000) to 21 ($200,000 and over), 6 corresponds to $50,000 to $59,000 and 9 corresponds to $80,000 to $89,999 a year

[2]Education was an ordinal variable ranging from 1 (below primary school) to 10 (doctoral degree), 6 corresponds to "some college"

**Table 3. Linear regression testing for main effects of political conservatism and conservative regional climate on vaccine hesitancy and booster hesitancy, United States, 2023.**

| Variable | Vaccine Hesitancy (n = 664) | | | Booster Hesitancy (n = 664) | | |
|---|---|---|---|---|---|---|
| | β | 95% CI | p | β | 95% CI | p |
| Political Conservatism | .22 | .16,.28 | <.001 | .18 | .11,.24 | <.001 |
| Conservative Political Climate | .11 | .05,.18 | .001 | .11 | .04,.19 | .002 |

Note: Both models controlling for age, gender, income, education, race, and vaccine access

together, our findings indicate that political climate, as represented through regional 2020 presidential election votes, has a stronger association with vaccine and booster hesitancy for individuals who are higher on political conservatism as compared to those who are politically liberal. This same trend was observed for self-reported booster status, but not self-reported vaccine status. Most strikingly, for booster status the interaction between political orientation and political climate indicated that the most liberal participants were over 65% likely to get boosted regardless of climate. The most conservative participants in liberal climates were 60% likely to get boosted, while conservatives in conservative climates were less than 35% likely to get boosted (OR =.83, p = .026).

**Table 4. Linear regression testing for main effects of and the interaction between political conservatism and conservative regional climate on vaccine hesitancy and booster hesitancy, United States, 2023.**

| Variable | Vaccine Hesitancy (n = 664) | | | Booster Hesitancy (n = 664) | | |
|---|---|---|---|---|---|---|
| | β | 95% CI | p | β | 95% CI | p |
| Political Conservatism | .23 | .23,.35 | <.001 | .18 | .12,.25 | <.001 |
| Conservative Political Climate | .11 | .03,.17 | .002 | .11 | .04,.18 | .003 |
| Conservatism * Conservative Climate | .09 | .00,.13 | .003 | .11 | .04,.17 | .003 |

Note: Both models controlling for age, gender, income, education, race, and vaccine access.

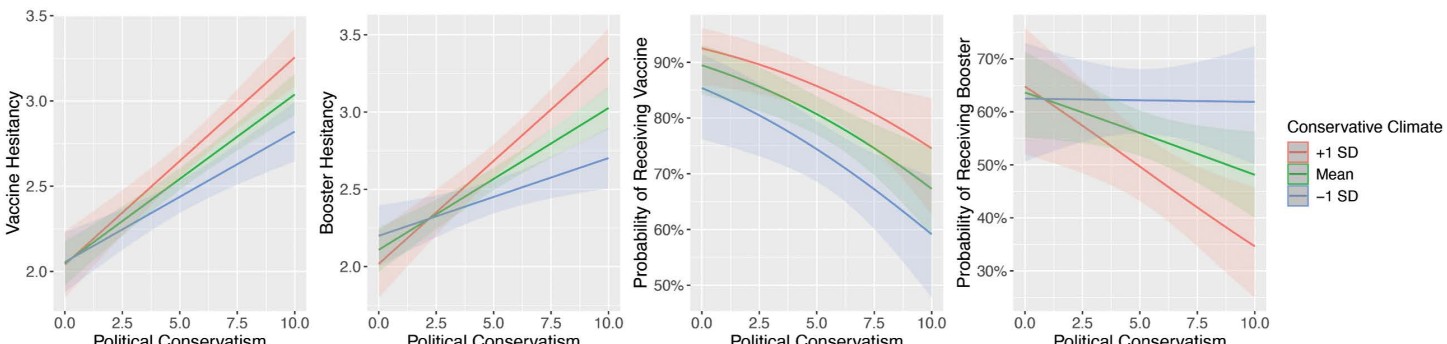

**Fig 2. Two-way interactions between political conservatism and conservative regional climate for vaccine hesitancy (N = 664), booster hesitancy (N = 664), vaccine status (N = 663) and booster status (N = 655), controlling for age, gender, education, income, race, vaccine access, and main effects of political conservatism and conservative political climate.** Four graphs depicting interactions between political conservatism and regional political climate for booster hesitancy, vaccine status, and booster status.

**Table 5. Logistic regression for main effects of political conservatism and conservative regional climate on vaccine status and booster status, United States, 2023.**

| Variable | Vaccine Status (n = 664) | | | Booster Status (n = 664) | | |
|---|---|---|---|---|---|---|
| | OR | 95% CI | p | OR | 95% CI | p |
| Political Conservatism | .68 | .54,.84 | <.001 | .83 | .69,.99 | .034 |
| Conservative Political Climate | .73 | .58,.90 | .004 | .76 | .63,.92 | .002 |

Note: Both models controlling for age, gender, income, education, race, and vaccine access.

**Table 6. Logistic regression testing for main effects of and the interaction between political conservatism and conservative regional climate on vaccine stauts and booster status, United States, 2023.**

| Variable | Vaccine Status (n = 664) | | | Booster Status (n = 664) | | |
|---|---|---|---|---|---|---|
| | OR | 95% CI | p | OR | 95% CI | p |
| Political Conservatism | .68 | .54,.84 | <.001 | .82 | .69,.98 | .022 |
| Conservative Political Climate | .72 | .58,.90 | .004 | .77 | .63,.93 | .004 |
| Conservatism * Conservative Climate | 1.02 | .82, 1.27 | .833 | .83 | .69,.99 | .026 |

Note: Both models controlling for age, gender, income, education, race, and vaccine access.

Political conservatism and conservative regional climate were both associated with higher levels of COVID-19 vaccine and booster hesitancy and lower rates of self-reported vaccination and boosting, when controlling for age, education, income, and race. Additionally, there was a significant interaction between political conservatism and conservative regional climate for vaccine hesitancy, booster hesitancy, and booster status. The interaction for vaccine hesitancy indicates that at the lowest levels of conservatism (i.e., liberalism) vaccine hesitancy is lower regardless of regional climate. As political conservatism increases vaccine hesitancy also increases, but this increase is even greater for individuals who live in a more conservative regional climate. There was a similar pattern for booster hesitancy, with more politically conservative individuals in a conservative regional climate having higher booster hesitancy than conservative individuals in a liberal regional climate. For the most politically liberal individuals, booster hesitancy was slightly lower in more conservative regional climates. The interaction for self-reported booster status was consistent with those for vaccine and booster hesitancy. Booster status decreased in likelihood as political conservatism increased. There was no difference in the likelihood of reporting receiving a booster based on conservative regional climate for the most politically liberal participants, but as political conservatism increased, the likelihood of receiving a booster was lower for participants in a conservative regional climate as compared to those in a liberal regional climate. The interaction for vaccination was not significant, this difference between attitudes (hesitancy) and action (vaccination), likely reflects the high prevalence of COVID-19 vaccine mandates. Our research is consistent with previous findings that conservative political orientation and a conservative political climate are both associated with COVID-19 vaccine hesitancy and decreased likelihood of receiving a COVID-19 vaccine [6,7,10,19]. Previous studies have focused on either political orientation or political climate. By looking at political orientation and region climate, we identify a novel and important finding, specifically that conservative individuals are more likely to received booster doses in liberal regions. While we did not asses the mechanism through which this occurs, these findings suggest that environmental factors (e.g., social pressure) may impact decisions to receive annual COVID-19 doses.

Because we used 2020 presidential election votes by zip code as a proxy for political climate, it must be noted that other zip code-based differences could be driving the associations noted in this paper. Regional factors (e.g., population density) may play a role in linking 2020 election votes to vaccine attitudes and behaviors. While our analyses controlled for demographic covariates commonly associated with vaccine hesitancy, we did not formally assess effect modification by these factors. Consequently, the reported pooled associations should be interpreted as average effects across the sample, and we cannot exclude the possibility that heterogeneity across demographic subgroups could produce aggregation effects such as Simpson's paradox. Another limitation in this dataset is the reliance on self-reported vaccine status. Self-report is susceptible to social desirability [30] and recall bias [31], and we cannot assume that reported vaccine status reflects actual vaccine status. Additionally, our sample was recruited through an online platform and only includes participants with access to this platform. Further, our sample includes only White and Black participants recruited through Qualtrics Panels, and may not generalize to broader populations in the United States. Finally, it should be noted that this data is cross-sectional and we cannot assume causal or directional associations. Despite these limitations, this research provides novel evidence that political climate in the US is associated with vaccine hesitancy and uptake for politically conservative individuals.

Identifying the specific mechanisms through which this happens would have significant implications for increasing vaccine uptake among politically conservative individuals across the US. The interaction between political climate and orientation suggests that liberal environments may exert more influence on conservative individuals than conservative environments exert on liberal individuals. It could also be the case that conservative individuals are more influenced by their environments than their liberal counterparts. Future research should explore the extent to which political climate, and other related regional variables, are associated with vaccine attitudes and behavior. Regardless, these findings are critical to our understanding of the association between political orientation on vaccine hesitancy. These results have significant implications for populations to target to increase COVID-19 vaccine uptake as we move towards annual vaccinations.

## Supporting information

**S1 File. Items from Life Experiences and COVID-19 – Qualtrics.**
(DOCX)

**S2 File. Confirmatory Factor Analysis.**
(DOCX)

**S3 Table. Participant Distribution by State.**
(DOCX)

**S4 File. Results for All Regression Models with Control Variables.**
(DOCX)

## Author contributions

**Conceptualization:** Rachel E. Dinero, William B. Monti.

**Data curation:** Rachel E. Dinero, William B. Monti.

**Formal analysis:** Rachel E. Dinero, William B. Monti.

**Funding acquisition:** Rachel E. Dinero, Brittany L. Kmush.

**Investigation:** Rachel E. Dinero.

**Methodology:** Rachel E. Dinero, William B. Monti.

**Project administration:** Rachel E. Dinero.

**Resources:** Rachel E. Dinero, William B. Monti.

**Software:** Rachel E. Dinero, William B. Monti.

**Supervision:** Rachel E. Dinero.

**Validation:** Rachel E. Dinero, Brittany L. Kmush.

**Visualization:** Rachel E. Dinero.

**Writing – original draft:** William B. Monti.

**Writing – review & editing:** Rachel E. Dinero, William B. Monti, Brittany L. Kmush.

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
