## [Decision Letter · Decision Letter 0]

17 Sep 2025

Dear Dr. Kmush,

Thank you for submitting your manuscript to PLOS ONE. After careful consideration, we feel that it has merit but does not fully meet PLOS ONE’s publication criteria as it currently stands. Therefore, we invite you to submit a revised version of the manuscript that addresses the points raised during the review process.

We look forward to receiving your revised manuscript.

Kind regards,

Mickael Essouma, M. D.

Academic Editor

PLOS ONE

Journal Requirements:

This research was funded by a Public Affairs and Policy Research Initiative Grant from Colgate University.

4. We noted in your submission details that a portion of your manuscript may have been presented or published elsewhere. Please clarify whether this publication was peer-reviewed and formally published. If this work was previously peer-reviewed and published, in the cover letter please provide the reason that this work does not constitute dual publication and should be included in the current manuscript.

Additional Editor Comments:

I. Major comments

-About the Introduction. I find it quite long. In my opinion, you would better start by highlighting the burden of COVID-19 in USA up to the end of the study period (in terms of morbidity and case fatality), a brief history of the introduction of COVID-19 vaccination in USA, and how COVID-19 vaccination influenced the burden of COVID-19 in USA based on the most compelling evidence; while also highlighting the scientific controversy surrounding the efficacy of COVID-19 since their approval (see Monash Bioeth Rev. 2024 Jun;42(1):28-54. doi: 10.1007/s40592-024-00189-z and https://jamanetwork.com/journals/jama-health-forum/fullarticle/2836434). You could then transition to a second paragraph to show that the controversy spread beyond the scientific milieu to reach the politics and the public, providing the overall viewpoint of the public and political agents about COVID-19 vaccine. This is where the authors would provide updated knowledge and knowledge gaps about the contribution of politics (notably political conservatism) to COVID-19 vaccine hesitancy among them and among the entire US population, and would describe how conservatist political ideologies, political climate and the current US president help shaped the narrative and trust around COVID-19 vaccine efficacy in political milieus and in the general US population. This would help the authors usher into the overarching goal of this study and the aim of the study taking into consideration the fact that COVID-19 vaccine efficacy or their life-saving potential is still being actively discussed among health community members , political agents and the general public. With editing efforts, this could lead to a-1.5 page introduction.

-About the Materials and Methods section. I suggest starting this section with a sub-section termed «Study design» where you would state that this was «an online behavioral study using the Qualtrics panel». After that, you would explicitly state that you conducted and reported this study according to a guide for psychological research based on data panel (current ref 26). This would help better understand the rest of the section and enhance the credibility of the study report. I am unable to access ref 26 full text and so, I do not know whether there is a checklist in that guide that you should eventually consider to fill and upload as supplemental Material to showcase the compliance with ref 26 guidelines. In line 85, you stated that participants were recruited from USA and zip codes seem to have been important variables when going through the PROCEDURES sub-section. Could you then go into more details with the geographic description of participants' living places in line 85: specific states where the study was conducted, in urban vs rural areas, in which zip code area, while explaining what zip codes represent in USA and perhaps their corresponding variables outside USA for international readers? Along with the suggestion to describe Qualtrix panels when mentioning the study design, consider completing the information in the sentence "Qualtrix panel...income" (lines 87 and 88) with specification in the PARTICIPANTS sub-section of further explicit information about the inclusion and exclusion criteria which are somewhat unclear in lines 88-94? For instance, what age, sex and gender groups did you include? What political ideology? What about COVID-19 vaccination status? Did you exclude individuals with neurocognitive deficiencies given that they may affect data quality (see "Data quality of platforms and panels for online behavioral research | Behavior Research Methods" https://link.springer.com/article/10.3758/s13428-021-01694-3)? Why did you include only two racial groups? What about participants'ethnic groups? What guided the choice of the income limit for participants? Did you include some zip codes to target particular economic groups of participants? What sampling method did you use for participants selection? How did you estimate the sample size? Could you add a "Data items or Variables" sub-section in between the PARTICIPANTS and PROCEDURES sub-sections to clearly describe the Qualtrix panel data used in this study with the variables (e.g., participants' sociodemographic and economic variables, political conservatism and its modalities, political climate and its modalities, and COVID-19 vaccine variables)? That description needs to be accompanied by a QUALTRIX questionnaire template used uploaded as a supplemental material. Consider reorganizing the current PROCEDURES sub-section to more clearly and deeply describe each of the research procedures carried out, except for statistical analysis which is described in a specific sub-section after the PROCEDURES sub-section: protocol preparation and registration, getting ethical matters sorted, participant enrolment (including the spread of questionnaires online, information of the target population and getting participants' responses), collection of survey responses/data, data storage before analysis (and eventually exportation towards the statistical analysis software). Could you complete the sentence in lines 96 and 97 with the IRB reference number? Given the inclusion of only two racial groups, can you claim to have complied with the recommendation for diversity and inclusivity when selecting study participants as mentioned in the 2024 Declaration of Helsinki? See line 97. This issue should be mentioned in the limitations statement of the Discussion section when commenting on the generalizability of study findings. The sentence in lines 98-100 would better be moved to the PARTICIPANTS sub-section, and the study period would better be mentioned at the beginning of the Materials and Methods section. I suggest adding a sub-section for the definition of operational terms (political conservatism, political climate, and COVID-19 vaccine terms) in between the PROCEDURES and STATISTICAL ANALYSIS sub-sections. The STATISTICAL ANALYSIS sub-section also needs more precision about data curation and quality control. Could you be more specific about the type of logistic regression performed for the main association analysed? What were the independent and dependent variables? How did you select the covariates? Along this line, it seems that keeping selecting race as a covariate (as did the authors) in association analyses could contribute to persistence of racial inequities in social (notably health) sciences research: see "“We adjusted for race”: now what? A systematic review of utilization and reporting of race in American Journal of Epidemiology and Epidemiology, 2020–2021 | Epidemiologic Reviews | Oxford Academic" https://academic.oup.com/epirev/article/45/1/15/7288093?login=false. Did you perform sensitivity analyses? Could you clearly explain how the moderation analysis mentioned in the manuscript's title was performed and why moderation analysis was selected over mediation analysis? How did you determine the effect magnitude estimates and their 95 percent confidence intervals? Why did you use both PR and OR as effect magnitude estimates? How do you present data in the text and in illustrations (e.g. how were tables and figures drawn, stratified/unstratified)? The sentence "For all analyses, ...from the analysis." in lines 147 and 148 belongs to the PARTICIPANTS sub-section.

-About the Results section. It is difficult to read, with no sub-section. I suggest the first sub-section on participants' social, demographic and economic characteristics. The second sub-section should correspond to the association assessed, and the third one to results of the moderation analysis. Could you highlight the process of participant selection in a figure?

-About the Discussion section. Consider first summarizing the results, then interpreting them, then the limitations and strengths statements, and finally, recommendations for policy makers, researchers, and clinicians.

II. Minor comments

-Amend the abstract, keywords, conclusion and references as required, and conform to PLOS One author guidelines when formatting the revised manuscript.

Reviewers' comments:

Reviewer's Responses to Questions

**Comments to the Author**

1. Is the manuscript technically sound, and do the data support the conclusions?

Reviewer #1: Yes

Reviewer #2: No

Reviewer #3: Yes

Reviewer #4: Partly

Reviewer #5: Partly

2. Has the statistical analysis been performed appropriately and rigorously?

Reviewer #1: Yes

Reviewer #2: No

Reviewer #3: Yes

Reviewer #4: Yes

Reviewer #5: No

3. Have the authors made all data underlying the findings in their manuscript fully available?

Reviewer #1: Yes

Reviewer #2: Yes

Reviewer #3: Yes

Reviewer #4: Yes

Reviewer #5: Yes

4. Is the manuscript presented in an intelligible fashion and written in standard English?

Reviewer #1: Yes

Reviewer #2: Yes

Reviewer #3: Yes

Reviewer #4: Yes

Reviewer #5: Yes

Reviewer #1: Dear Author(s),

This manuscript presents a timely and original investigation. The study is methodologically rigorous, with clearly defined research objectives, appropriate statistical techniques, and a transparent presentation of results. The analyses are carefully executed and align with the study’s stated hypotheses, with the findings logically supporting the conclusions.

The introduction does a commendable job of situating the study within the existing literature, but the framing could be strengthened by more explicitly highlighting the novelty of this work in comparison to prior studies. While the discussion section interprets the results appropriately, it could benefit from more direct linkage to potential practical implications and a clearer acknowledgment of the limitations beyond those already noted. This would help readers contextualize the findings for both academic and applied settings.

The statistical methods are robust and well-documented. It is particularly positive that the data are made fully available, ensuring reproducibility. Figures and tables are clear, relevant, and appropriately labeled, though a few captions could be expanded to ensure they are fully interpretable without referring back to the main text. The manuscript is written in clear, standard English, with only a few minor grammatical refinements needed to further enhance readability.

In short, this is a solid, technically competent study that advances understanding in its domain. The suggested revisions are relatively minor and focus on enhancing clarity, framing, and interpretive depth rather than making substantive analytical changes.

Best regards,

Reviewer #2: I. SUMMARY

This manuscript examines the relationship between individual political orientation, local political climate, and COVID-19 vaccine hesitancy. While the topic is of potential public health and sociopolitical interest, the manuscript suffers from fundamental flaws in conceptual framing, theoretical grounding, and scientific value. These weaknesses render its conclusions essentially meaningless, regardless of what the statistical output happens to show.

1. Premise and Scientific Merit

The central premise — that political orientation and political climate are associated with COVID-19 vaccine hesitancy — is already well-documented to the point of redundancy. This work does not substantially advance that knowledge. Instead, it repackages an established correlation without offering:

- New explanatory mechanisms that could be tested or falsified.

- Nuanced evaluation of why hesitancy exists in various populations.

- Consideration of whether the hesitancy might be rational under certain conditions.

A central tenet of scientific inquiry is that the question itself must have potential to generate meaningful or actionable insight. This study does not meet that threshold; even if flawlessly executed, it could not produce findings that alter scientific understanding or public health practice.

2. Lack of Engagement with Intervention Efficacy

The manuscript treats COVID-19 vaccination as a universally optimal intervention and implicitly assumes that increased uptake is inherently desirable. There is no attempt to evaluate the real-world efficacy, safety, or cost-benefit profile of the intervention in question at the time of data collection. This omission is not merely a gap in discussion — it invalidates the interpretive frame. Without assessing whether the behavior being measured is objectively beneficial, the act of labeling hesitancy as problematic is scientifically hollow.

3. Theoretical Weakness

The introduction attempts to invoke concepts such as group polarization and resistance to persuasion, but these are deployed as rhetorical devices rather than as elements of a coherent, testable model. There is no operational definition of how these mechanisms would be detected, nor any consideration of alternative models that could explain the same data. The result is a literature review in service of a foregone conclusion, rather than a genuine hypothesis-building exercise.

4. Methodological Concerns

Measurement validity: The political climate variable is a blunt instrument (% Republican vote by ZIP code) that may capture a variety of demographic, cultural, and socioeconomic factors unrelated to political ideology per se. These are neither acknowledged nor controlled for.

Scale modification: The Attitudes towards Adult Vaccination Scale was altered for COVID-specific use without any psychometric re-validation, raising concerns about reliability and construct validity.

Exclusion handling: Participant attrition is reported piecemeal, making it difficult to follow the data pipeline from initial sample to final analytic N.

5. Interpretive Overreach

Correlational results are treated as if they illuminate underlying cognitive or epistemic deficiencies in certain political groups. No evidence is provided to support such inferences, and alternative explanations (e.g., differing trust in institutions, differential access, or varying personal risk assessments) are not meaningfully considered. This transforms the manuscript from a descriptive study into an exercise in political attribution, which is outside the scope of empirical demonstration here.

6. Contribution to the Field

The manuscript’s contribution is minimal. It offers no novel methodology, no theoretical innovation, and no actionable public health insight. Similar results have been repeatedly reported in both academic and popular media outlets since 2020. In its current form, this work risks functioning more as political commentary than as science.

II. DETAILED REVIEW

A) INTRO

"In the US, political conservatism has been consistently associated with greater COVID-19 vaccine hesitancy (1–7) and lower COVID-19 vaccine uptake (5,8,9)"

Comment: Opening with an already well-established, heavily publicized correlation signals this study is unlikely to produce novel insight. If the field already accepts this as a settled empirical fact, the manuscript must immediately justify why retesting it is scientifically useful. No such justification follows.

"While political conservatism has been historically associated with lower trust in the scientific community..."

Comment: This “historically associated” claim is sweeping and imprecise. No operational definition of “trust in the scientific community” is given here, and the citations do not establish causality. This language sets up a simplistic cause-effect chain without acknowledging other mediating variables.

"...the increased partisan divide in vaccine hesitancy may be attributed, at least in part, to the politicization of COVID-19 during the Trump administration (12)"

Comment: This is a politically charged attribution offered without empirical demonstration in this study. Even if true, it is a claim about historical causation that cannot be substantiated by the present dataset.

"President Donald Trump initially downplayed the COVID-19 pandemic, even labeling it as a hoax (13)..."

Comment: This paragraph functions rhetorically, not scientifically. The detail is not necessary to establish the study’s variables, and its inclusion risks signaling partisan bias. Its presence here does nothing to clarify hypotheses or inform operationalization.

"Nonetheless, the influence of political orientation on vaccine hesitancy has been increasing..."

Comment: Again, this reiterates the opening claim. The intro has now made the same “conservatives = more hesitant” point three times in different words, with no additional conceptual depth. This redundancy inflates length without adding value.

"There is an emerging body of evidence that the broader political climate..."

Comment: This is a potentially interesting angle (regional climate effects) but is described in the same correlational terms as before. No competing hypotheses are introduced, no mechanisms proposed beyond “politics affects attitudes,” and no argument for why this is worth re-examining in 2023.

"While previous research has identified the unique impact of both political orientation and political climate...little research on the interaction..."

Comment: This should have been the opening sentence. It is the closest the introduction comes to stating a gap in the literature — but it arrives buried after several paragraphs of repetitive background and political editorializing.

"Group polarization...could lead those who are politically conservative and live in politically conservative areas to have even stronger vaccine hesitancy..."

Comment: This is a generic invocation of a social psychology concept without operational definition. How is “group polarization” measured here? How does the design distinguish polarization from simple majority influence? This is theory-name-dropping, not hypothesis development.

"For example, self-identified Republicans reported greater intention to vaccinate after watching a video of Trump..."

Comment: The example is selectively chosen to reinforce the prior partisan framing. It is anecdotal in the context of the argument and does not advance the operationalization of the current study’s variables.

"In this way, when conservative individuals...the group polarization effect is created..."

Comment: This is an assumption stated as fact. The present study does not measure interpersonal influence, exposure to like-minded views, or any group-level dynamics. “Group polarization” is being asserted as the mechanism without evidence.

"Individuals who are exposed to positive vaccine messaging...we might expect that conservative individuals who live in liberal areas..."

Comment: The shift here from conservative-in-conservative-areas to conservative-in-liberal-areas feels unfocused. The introduction now contains multiple, partially contradictory speculations without clearly stating the testable predictions or expected interaction patterns.

"Alternatively, resistance to persuasion is a well-documented psychological phenomenon..."

Comment: This is a second, competing explanatory frame, also undeveloped. The introduction now lists two incompatible mechanisms without specifying which is predicted to dominate or under what conditions.

"Here, we examined the moderating role of political climate on the association between individual political orientation and COVID-19 vaccine hesitancy."

Comment: This is the first clear statement of the research aim — and it appears at the end of the introduction. By this point the reader has waded through redundant background and partisan color commentary that could have been condensed into three sentences.

B) METHODS

i) Participants

"Participants were recruited from the United States by Qualtrics Panels(26)"

Comment: Opt-in online panel samples are inherently self-selected and not probability-based. This is fine for exploratory work but disqualifies the study from making strong generalizations about “Americans” as a whole. No acknowledgement of this limitation appears here.

"The data and analysis presented in this paper are part of a larger project..."

Comment: This raises immediate concerns about data mining. Without a preregistered analysis plan, the “larger project” could have tested multiple variables, cherry-picking the political–vaccine link for publication. No declaration of how this study was scoped relative to the larger dataset.

"Qualtrics Panels recruited participants across four quota groups based on race and income..."

Comment: The quota sampling is not stratified random sampling — it’s cosmetic demographic balancing. The chosen quotas (White/Black, above/below median income) are arbitrary with respect to the stated research question and introduce unnecessary complexity that is never analytically leveraged.

"...no more than 60% of the participants be of any gender."

Comment: Again, quota design is arbitrary. Gender balancing is fine, but this appears to be a procedural checkbox, not a hypothesis-driven decision.

Procedures

"Political conservatism was assessed across two items... rated 0 to 10... averaged to form a political conservatism scale."

Comment: The measurement is narrow — “conservatism” is inferred from only two self-report sliders on social and economic issues. No psychometric validation is cited for collapsing these into a single score, nor is the correlation between the two items reported. This invites construct validity concerns.

"Conservative regional climate... calculated from 2020 presidential election results... % Republican vote by ZIP code."

Comment: This variable is a blunt proxy. Political climate is collapsed into a single presidential vote share metric, ignoring voter turnout rates, third-party votes, temporal changes between 2020 and survey collection in 2023, issue-specific political leanings that might diverge from presidential voting. As well, no control for confounding geographic factors (urban/rural, education levels, local COVID policy intensity) is attempted, making causal inference impossible.

"In cases where a zip code included multiple FIPS codes, data was averaged..."

Comment: Averaging election returns across multiple FIPS codes further dilutes the precision of the “political climate” measure. This method produces an ecological variable with unknown reliability.

ii) Vaccine and Booster Measures

"Self-reported vaccine status..."

Comment: Self-report is fine for some measures, but no attempt is made to validate these reports or assess recall/social desirability bias. Given the politicization of the topic, measurement error here is a real threat.

"Vaccine hesitancy was measured using the seven items from the Attitudes towards Adult Vaccination Scale"

Comment: Major validity problem — the scale was altered (wording changed to COVID-specific) with no psychometric re-validation. The authors cannot assume reliability or factor structure remains intact post-modification. This is particularly problematic given they use the scale as a primary outcome.

iii) Statistical Analyses

"Pearson correlations... t-tests... logistic regression models... controlling for age, gender, education, income, and race."

Comment: The control variables are minimal and do not include plausible covariates such as local COVID rates, occupation type, comorbidities, or trust in specific institutions. The omission of these makes any claim of “political orientation causes hesitancy” statistically meaningless.

"Any participants with missing data on any of these variables were excluded from the analysis."

Comment: This is listwise deletion without justification. No exploration of whether missingness is random, nor any multiple imputation approach. This risks biasing the sample further toward respondents with complete, possibly non-representative data.

"Two-sided p-values less than 0.05 were considered statistically significant."

Comment: Standard threshold — but there’s no discussion of multiple comparison correction despite running numerous models and correlation tests. This inflates Type I error risk, meaning “significant” results could be statistical noise.

C) RESULTS

"Qualtrics Panels collected data from 1777 participants, and eliminated 970 participants... resulting in a sample of 798 validated participants."

Comment: Losing over half the initial sample to attention check failures or incomplete surveys is a bright red flag for data quality. This isn’t just “cleaning” — it’s an indication that the recruitment pool was not engaged or representative. Attrition on this scale guts external validity.

"We eliminated an additional 115 participants who did not provide valid zip code data or whose zip code did not have corresponding election data."

Comment: That’s another ~14% gone. You’re now down to 38% of the original sample. At this point, you’ve essentially self-selected for respondents willing to fully disclose location data and complete a long survey — exactly the sort of filtering that will exacerbate ideological skew. No discussion of this bias appears.

"Participants ranged in age from 18 to 94 years"

Comment: No indication of how age distribution compares to the general US population — again, no representativeness check.

"519 of participants reported receiving the initial COVID-19 vaccine and 365 reported receiving at least one COVID-19 booster."

Comment: These uptake rates are far lower than CDC national estimates at comparable time points, suggesting a non-representative sample. Without weighting or adjustment, any national-level inferences are invalid.

"Both vaccine hesitancy and booster hesitancy scores ranged from 1 to 5..."

Comment: They report range, mean, SD — but still no check of scale reliability after modification. Without that, these are just arbitrary composites of unvalidated items.

"the % Republican vote by zip code ranged from..."

Comment: Reporting range and SD doesn’t fix the problem that this metric is stale (from 2020) and ecologically coarse.

"Vaccine hesitancy was positively correlated with booster hesitancy"

Comment: r = .86 simply means the two “different” hesitancy scales are basically measuring the same construct — not surprising since they’re word-substitution versions of the same unvalidated instrument. This is tautology masquerading as insight.

"...political conservatism and conservative regional climate"

Comment: r = .36 is a modest correlation; r = .16 is trivially small. With N = 683, even trivial associations will be “statistically significant” — but they’re explaining barely 2–13% of the variance. These are the sorts of effects that evaporate in better-controlled designs.

"Unvaccinated participants were significantly younger... "

Comment: This is pure confirmation of what was baked into the premise. The model is underspecified — “political conservatism” is correlated with multiple sociodemographics here, any of which could be the actual driver. Without multivariate disentangling of collinearity, these t-test “findings” are little more than descriptive stereotypes.

"Similarly, participants who reported not receiving a booster..."

Comment: This is a repeat of the vaccine status pattern, just with boosters. Again, no attempt to adjust for overlapping covariates beyond a few controls in later regressions. The paper is making the same point twice and treating it as separate evidence.

Everything here is correlational and ecologically confounded. Yet the narrative implicitly treats “living in a conservative climate” as an independent driver of hesitancy, without testing for other area-level variables (e.g. education rates, healthcare access, prior infection rates), or establishing temporal ordering (i.e. did political climate cause hesitancy, or did both stem from a third factor?).

What the data actually shows: in an opt-in, quota-balanced panel surveyed in spring 2023, self-reported COVID vaccination and booster uptake correlate modestly with a 2-item conservatism index and trivially with 2020 presidential vote share aggregated to ZIP codes.

What the data does not show: causation, mechanisms (polarization/persuasion), efficacy of any intervention, or meaningful predictive power beyond generic demographics.

Nothing actionable can be inferred from this work without further experiments, better measures, and serious controls.

III. CONCLUSION

The manuscript’s central premise — that political orientation and local political climate are associated with COVID-19 vaccine hesitancy — is already well established in both the academic and popular literature. The study offers no new theoretical model, explanatory mechanism, or intervention test. Even if the analyses were flawless, the research question is scientifically low-yield and cannot advance understanding or public health practice.

Sampling included arbitrary quotas that are unrelated to the stated research aim. Attrition was extreme, raising concerns about bias and representativeness. Key measures suffer from construct validity issues. Control variables omit plausible confounders (local COVID burden, healthcare access, comorbidities, trust in public health). Missing data were handled by listwise deletion with no missingness analysis. No multiple-comparison correction was applied despite numerous tests.

The findings are predictable from the operational definitions: modest correlation between self-reported conservatism and hesitancy and trivial correlation with political climate. Booster hesitancy and vaccine hesitancy correlation indicates redundancy rather than independent confirmation. Attrition and non-probability sampling preclude generalization; no weighting or sensitivity analyses are reported. The “climate” effects are statistically significant only due to sample size and would likely vanish under stronger controls.

The Discussion implicitly treats observed correlations as evidence of causation and invokes “group polarization” without having measured any mechanism (social networks, media exposure, interpersonal influence). Causal language is applied to variables measured cross-sectionally, ecologically, and with unvalidated instruments. Hesitancy is framed as inherently problematic without assessing the contemporaneous efficacy/risk profile of the intervention (vaccination in early 2023), reducing the interpretation to a value judgment rather than a scientific conclusion.

Policy implications are speculative and unsupported — no interventions were tested, yet recommendations for targeted messaging are offered. Sampling and measurement limitations are not adequately engaged, and effect sizes are ignored in favor of p-value significance.

Reviewer #3: The manuscript examines a salient moderating role of regional political climate between self-reported political tendency and vaccine (or booster) hesitancy. I would like to kindly suggest several robustness-checks on the key variable of this study, regional political climate. The author(s) carefully built the variable at the zip code level which is geographically granular, potentially reflecting on neighborhood-level political atmosphere. However, maybe a more macro-level data such as state can better capture political climate in which a respondent resides. So it would be great to use state-level 2020 election results instead of zip code-level data as a robustness check. This especially makes sense because the state government led most responses to the COVID-19 pandemic, rather than zip code-level local responses. In the same context, an additional table on where the 683 respondents live state by state will be informative and important for readers because some readers may raise a question of "what if most of the 683 respondents live in California? Or Texas?" An additional supplemental table may help clarify this natural question. More importantly, a basic question would be if and the extent to which political climate differs across zip code areas. Is there a huge variation across adjacent zip code areas within the same county or state? Or are they nearly same to each other within the same county or state? This evidence would be important when the author(s) justify WHY they use zip code level variable. Hope these comments be able to help improve the manuscript. Thank you very much.

Reviewer #4: I am writing to provide a critical evaluation of the manuscript entitled “Political Conservatism, Political Climate, and COVID-19 Vaccination Hesitancy and Uptake in the United States.” The paper analyzes a Qualtrics Panels survey administered in March–April 2023 to assess how local political climate conditions the association between individuals’ political conservatism and both COVID-19 vaccine hesitancy and booster uptake. The central empirical claim is that liberals are consistently less hesitant regardless of place, whereas conservatives report lower hesitancy and higher booster uptake when they reside in more liberal political climates. The authors estimate linear models for attitudinal indices and logistic models for uptake, focusing on a conservatism × local vote-share interaction while adjusting for standard demographics.

The question is important and timely. Since the first year of vaccine rollout, independent population-level sources have documented unusually strong associations between vaccination and partisan voting patterns, far exceeding analogous correlations for seasonal influenza; situating the manuscript explicitly against this backdrop will help readers interpret the magnitude and direction of the reported effects.

The paper’s contextual moderation claim also aligns with two complementary literatures: work showing that the political composition of interpersonal networks predicts vaccine confidence and behavior, and classic theories of social influence and cross-pressures indicating that prevailing local norms can dampen or amplify individual predispositions. In addition, experimental evidence has shown that elite partisan cues can increase vaccination intentions among Republicans, underscoring the plausibility of the mechanisms the authors invoke while also highlighting the difficulty of distinguishing interpersonal from elite informational channels in observational designs.

The data and design are generally appropriate for the questions posed, and the authors deserve credit for applying response-quality screens typical of online panel work. That said, two measurement decisions warrant revision. First, the hesitancy construct is adapted from existing adult-vaccination items but reworded for COVID-19. Best practice requires reporting fresh psychometric evidence when items are modified or redeployed to a new context. Following COSMIN guidance, the manuscript should document internal consistency (e.g., α or ω), item–total correlations, dimensionality (CFA/parallel analysis), and, given the paper’s focus, measurement invariance across key subgroups such as party identification. Second, the “political climate” measure is built by assigning 2020 presidential vote shares to respondents’ ZIP codes using crosswalks. Because USPS ZIPs are not stable statistical areas, and Census ZCTAs only approximate them, such crosswalks can induce nontrivial boundary and smoothing error. I recommend sensitivity analyses at alternative geographic units (ZCTA, county) and discussion of potential misclassification introduced by ZIP–ZCTA conflation.

The statistical specification is serviceable but can be improved in ways that would materially strengthen the inferences. First, because the key moderator is a place-based attribute, standard errors should be clustered geographically or, better, modeled with multilevel structure to reflect shared context and avoid overstated precision. A multilevel framework also enables partial pooling across places and principled estimation of cross-level interactions; the canonical reference remains Gelman and Hill.

Second, given the use of a nonprobability online panel, I encourage the authors to report weighted estimates or apply multilevel regression and post-stratification (MRP) to align sample margins to ACS or CPS benchmarks; both the AAPOR task force and subsequent methodological work have set clear expectations here. Third, because political climate is correlated with structural covariates—urbanicity, race composition, and socioeconomic status—the models should incorporate ecological controls (e.g., rural–urban continuum codes or vulnerability indices). CDC surveillance has consistently shown lower COVID-19 vaccination in rural counties, a pattern relevant to interpretation of any contextual moderation.

A few issues of clarity and transparency also deserve attention. The abstract and methods alternate between odds ratios and “PR,” suggesting a prevalence-ratio estimator for booster outcomes; if Poisson regression with robust variance was used, this must be stated explicitly and applied consistently, with 95% confidence intervals for all effect measures. The moderation results would benefit from covariate-adjusted marginal-effects plots, presented alongside the distribution of the moderator to avoid extrapolation beyond observed ranges. Because the observed period corresponds to early uptake of the bivalent booster—when overall adult coverage remained modest and varied across groups—anchoring the manuscript’s descriptive statistics against CDC population estimates would enhance external validity and give readers a concrete sense of scale.

Interpretively, the manuscript’s conclusions are plausible and are framed in a manner consistent with the literatures noted above. Yet causal language should be tempered. Individuals do not randomly sort into political contexts; the interaction between ideology and place could reflect unmeasured selection or access channels (e.g., provider availability, mandate environments) rather than pure normative pressure. To probe this, the authors might add robustness checks using alternative geographies; introduce proxies for local information environments; and, if feasible, re-estimate the moderation in multilevel models that include area-level covariates. Finally, a brief “measurement appendix” documenting the exact wording of the adapted items, response options, and scale properties would meet contemporary reporting standards for patient-reported measures and reduce concerns about construct drift.

In sum, the manuscript addresses a consequential and theoretically rich question and presents evidence consistent with the notion that local political climates condition the effect of individual conservatism on vaccine hesitancy and uptake. With revisions that (i) document the adapted scale’s psychometrics in line with COSMIN recommendations; (ii) reconcile and standardize effect measures for booster outcomes; (iii) adopt clustered or multilevel inference and appropriate post-stratification or MRP for generalization; and (iv) demonstrate robustness to alternative spatial operationalizations of political climate while acknowledging ZIP/ZCTA limitations, the paper would make a valuable contribution to the study of politically charged health behaviors. Given the importance of these methodological and reporting improvements, my editorial recommendation is at least “minor revision.”

Reviewer #5: There is a lot to like about the manuscript. Vaccine hesitancy is an interesting topic that should have broad appeal. The article is well written, both because it is easy to read and because it is careful with language. The core finding is interesting: conservatives who live in places with more liberals are less hesitant to be vaccinated for COVID-19.

However, I have some serious reservations about the rigor of the science. To me, the study seems like it is better suited to be a piece of a research paper---not a stand alone project. There is too much data and information missing to make an inference like the authors claim. I cannot recommend publication, but I think more work to explore these relationships will develop the piece into a solid publication. My major concerns boil down to sampling, attrition, and measurement.

Sampling: The authors start out with a little less than 2000 adults in a their survey. The authors study persons who identify as only White, or only Black. In doing so they limit themselves to a majority population and one minority population, both of which are shrinking as a proportion of the United States. (See https://www.census.gov/library/stories/2021/08/improved-race-ethnicity-measures-reveal-united-states-population-much-more-multiracial.html) While they represent the majority of the population, they are a poor selection for understanding the difference between conservative and liberal parts of the United States. The sample likely represents conservatives much better than it does liberals. It is missing multi-racial persons, the fastest growing population in the United States, and its missing Hispanics and Asians. All three of these groups are more likely to identify as Democratic or lean Democratic (See https://www.pewresearch.org/politics/2024/04/09/partisanship-by-race-ethnicity-and-education/) and black Americans are poor proxies for these groups. Controlling does not help with the fact of all together missing 25% of the population.

Attrition: The authors lose two thirds of their survey population because they fail to answer attention questions correctly or key questions in the survey. This is missing data that may very well bias results. Given the magnitude it may make relationships that are significant seem to not be. The authors do not tell us if the sample still balances on income and race. The authors also do not tell us how attrition is related to their moderator or if it is spatially correlated. These relationships all need to be explored. Other data may be needed to clarify the impact of the unfortunately high attrition rate.

Measurement: The authors conflate living in a liberal area with living in an area where most people voted for Biden and living in a conservative area with living in an area where most people voted for Trump. While these are no doubt overlapping categories, it's not obvious that this voting snapshot doesn't account for the results. It may be that many of the 'conservatives' that live in Biden voting areas, actually live in areas that keep on switching. These places are hardly the same as the densely populated counties that comprise the majority of Democratic votes. Robustness is necessary. The authors might use other elections like gubernatorial elections or senate elections. They could also use measures like the moral values measures that Ben Enke uses.

A final note the FIPS-ZIP matching is a problem for the analysis. There will be one to one matching among one party's stronghold and averaging across the others. It throwing away variation in a skewed way.

**Do you want your identity to be public for this peer review?** For information about this choice, including consent withdrawal, please see our Privacy Policy

Reviewer #1: **Yes:** YUANHAO TIAN

Reviewer #2: No

Reviewer #3: No

Reviewer #4: No

Reviewer #5: No

---

## [Author Response · Author response to Decision Letter 1]

3 Nov 2025

PONE-D-25-30422

Regional political climate's moderating role in the association between political conservatism and COVID-19 vaccine hesitancy in the United States

PLOS ONE

Dear Dr. Kmush,

Thank you for submitting your manuscript to PLOS ONE. After careful consideration, we feel that it has merit but does not fully meet PLOS ONE’s publication criteria as it currently stands. Therefore, we invite you to submit a revised version of the manuscript that addresses the points raised during the review process.

We look forward to receiving your revised manuscript.

Kind regards,

Mickael Essouma, M. D.

Academic Editor

PLOS ONE

Journal Requirements:

Author Response: The file has been edited to match the style requirements.

This research was funded by a Public Affairs and Policy Research Initiative Grant from Colgate University.

Author Response: Statement added to financial disclosure statement

Author Response: Statements were removed from end of paper and all information is provided in Methods.

4. We noted in your submission details that a portion of your manuscript may have been presented or published elsewhere. Please clarify whether this publication was peer-reviewed and formally published. If this work was previously peer-reviewed and published, in the cover letter please provide the reason that this work does not constitute dual publication and should be included in the current manuscript.

Author Response: Our cover letter has been updated to reflect this.

Additional Editor Comments:

I. Major comments

-About the Introduction. I find it quite long. In my opinion, you would better start by highlighting the burden of COVID-19 in USA up to the end of the study period (in terms of morbidity and case fatality), a brief history of the introduction of COVID-19 vaccination in USA, and how COVID-19 vaccination influenced the burden of COVID-19 in USA based on the most compelling evidence; while also highlighting the scientific controversy surrounding the efficacy of COVID-19 since their approval (see Monash Bioeth Rev. 2024 Jun;42(1):28-54. doi: 10.1007/s40592-024-00189-z and https://jamanetwork.com/journals/jama-health-forum/fullarticle/2836434). You could then transition to a second paragraph to show that the controversy spread beyond the scientific milieu to reach the politics and the public, providing the overall viewpoint of the public and political agents about COVID-19 vaccine. This is where the authors would provide updated knowledge and knowledge gaps about the contribution of politics (notably political conservatism) to COVID-19 vaccine hesitancy among them and among the entire US population, and would describe how conservatist political ideologies, political climate and the current US president help shaped the narrative and trust around COVID-19 vaccine efficacy in political milieus and in the general US population. This would help the authors usher into the overarching goal of this study and the aim of the study taking into consideration the fact that COVID-19 vaccine efficacy or their life-saving potential is still being actively discussed among health community members , political agents and the general public. With editing efforts, this could lead to a-1.5 page introduction.

Author Response: Thank you for this thoughtful suggestion and references. We have significantly edited the introduction to reflect these suggestions.

-About the Materials and Methods section. I suggest starting this section with a sub-section termed «Study design» where you would state that this was «an online behavioral study using the Qualtrics panel». After that, you would explicitly state that you conducted and reported this study according to a guide for psychological research based on data panel (current ref 26). This would help better understand the rest of the section and enhance the credibility of the study report. I am unable to access ref 26 full text and so, I do not know whether there is a checklist in that guide that you should eventually consider to fill and upload as supplemental Material to showcase the compliance with ref 26 guidelines.

Author Response: Thanks for these constructive suggestions. We added a PLOS One reference for increased accessibility. We also created a Participant Recruitment section and included a detailed description of Qualtrics Panels. We also included the section Attention Checks under Methods and Materials with a description of additional attention checks used in our survey (as recommended by original ref 26) to ensure quality data from respondents.

In line 85, you stated that participants were recruited from USA and zip codes seem to have been important variables when going through the PROCEDURES sub-section. Could you then go into more details with the geographic description of participants' living places in line 85: specific states where the study was conducted, in urban vs rural areas, in which zip code area, while explaining what zip codes represent in USA and perhaps their corresponding variables outside USA for international readers?

Author Response: We added a table with number of participants by state and included a description of zip codes in the Procedures sub-section. We do not currently have data on the population density or size of the zip codes.

Along with the suggestion to describe Qualtrix panels when mentioning the study design, consider completing the information in the sentence "Qualtrix panel...income" (lines 87 and 88) with specification in the PARTICIPANTS sub-section of further explicit information about the inclusion and exclusion criteria which are somewhat unclear in lines 88-94? For instance, what age, sex and gender groups did you include? What political ideology? What about COVID-19 vaccination status? Did you exclude individuals with neurocognitive deficiencies given that they may affect data quality (see "Data quality of platforms and panels for online behavioral research | Behavior Research Methods" https://link.springer.com/article/10.3758/s13428-021-01694-3)?

Author Response: We added content to the Participant Recruitment section and added the Attention Check section to address these important questions.

Why did you include only two racial groups? What about participants'ethnic groups? What guided the choice of the income limit for participants? Did you include some zip codes to target particular economic groups of participants? What sampling method did you use for participants selection? How did you estimate the sample size?

Author Response: We clarified these important issues in the Study Design section, indicating that beyond the race and income quota groups, which were used for the larger project from which this data is extracted, there were no exclusions. Sample size was based on funding available for this project. Our goal was to recruit the largest number of participant possible within our budget. All participant compensation is managed through Qualtrics Panels, with the researcher paying a flat fee for the recruitment of the sample size. Given the planned analyses, we could have recruited fewer participants and still had statistical power, but hoped to maximize our funding and recruit a larger sample.

Could you add a "Data items or Variables" sub-section in between the PARTICIPANTS and PROCEDURES sub-sections to clearly describe the Qualtrix panel data used in this study with the variables (e.g., participants' sociodemographic and economic variables, political conservatism and its modalities, political climate and its modalities, and COVID-19 vaccine variables)? That description needs to be accompanied by a QUALTRIX questionnaire template used uploaded as a supplemental material.

Author Response: We created the Variables heading and added the study materials to the supplement from the online repository.

Consider reorganizing the current PROCEDURES sub-section to more clearly and deeply describe each of the research procedures carried out, except for statistical analysis which is described in a specific sub-section after the PROCEDURES sub-section: protocol preparation and registration, getting ethical matters sorted, participant enrolment (including the spread of questionnaires online, information of the target population and getting participants' responses), collection of survey responses/data, data storage before analysis (and eventually exportation towards the statistical analysis software).

Author Response: We reorganized the Procedures sub-sections to improve the clarity and readability of this section.

Could you complete the sentence in lines 96 and 97 with the IRB reference number?

Author Response: Added.

Given the inclusion of only two racial groups, can you claim to have complied with the recommendation for diversity and inclusivity when selecting study participants as mentioned in the 2024 Declaration of Helsinki? See line 97.

Author Response: Yes, this is an excellent question. The Declaration of Helsinki required justification for exclusion of participants based on demographic characteristics. The proposal for the initial study included a strong rationale, which was evaluated and approved by the Institutional Review Board. Because the goal of the study was to assess the unique influence of racial and economic marginalization, specific racial and economic groups were selected. The inclusion of White and Black participants allowed for the assessment of the unique contributions of race, income, and marginalization. These criteria were not relevant to the present analysis and certainly present a significant limitation that should be addressed in the limitations statement.

This issue should be mentioned in the limitations statement of the Discussion section when commenting on the generalizability of study findings.

Author Response: Absolutely, the statement in the limitations section reads, “Additionally, our sample was recruited through an online platform and only includes participants with access to this platform. Further, our sample includes only White and Black participants, and may not generalize beyond these demographics.”

The sentence in lines 98-100 would better be moved to the PARTICIPANTS sub-section, and the study period would better be mentioned at the beginning of the Materials and Methods section.

Author Response: The Materials and Methods section was reorganized to address these concerns.

I suggest adding a sub-section for the definition of operational terms (political conservatism, political climate, and COVID-19 vaccine terms) in between the PROCEDURES and STATISTICAL ANALYSIS sub-sections.

Author Response: We have updated to include the operational terms for these variables.

The STATISTICAL ANALYSIS sub-section also needs more precision about data curation and quality control.

Author Response: Content was added to Participant Recruitment address data curation and quality control.

Could you be more specific about the type of logistic regression performed for the main association analysed?

Author Response: We have updated the Statistical Analysis to specify regression types.

What were the independent and dependent variables? How did you select the covariates?

Author Response: Predictor and outcome variables are now identified in Study Design. Control variables were selected based on existing literature. They were included to determine the association between predictors and outcomes when controlling for known associations.

Along this line, it seems that keeping selecting race as a covariate (as did the authors) in association analyses could contribute to persistence of racial inequities in social (notably health) sciences research: see "“We adjusted for race”: now what? A systematic review of utilization and reporting of race in American Journal of Epidemiology and Epidemiology, 2020–2021 | Epidemiologic Reviews | Oxford Academic" https://academic.oup.com/epirev/article/45/1/15/7288093?login=false.

Author Response: This is an excellent point. Previous analysis of this data found race to be significantly associated with vaccine attitudes and behavior. Therefore, our goal in the present research was to explain variance in vaccine attitudes and behavior beyond that which we already know can be explained by race.

Did you perform sensitivity analyses?

Author Response: While we did not predict our findings to apply at the state level, we did run all models at the state level. While the other associations were consistent, the political climate variable was no longer significant. We did not predict that this analysis would be similar given the lack of granularity of climate at the state-level, but we could include this analysis at the editor’s discretion. In the submitted revision, we revised the vaccine and booster hesitancy scales consistent with the findings of our confirmatory factor analysis. Assessing the variables in this way did not alter the significant associations found in our predictor variables (i.e., political orientation, political climate, t

---

## [Decision Letter · Decision Letter 1]

25 Nov 2025

Dear Dr. Kmush,

Thank you for submitting your manuscript to PLOS ONE. After careful consideration, we feel that it has merit but does not fully meet PLOS ONE’s publication criteria as it currently stands. Therefore, we invite you to submit a revised version of the manuscript that addresses the points raised during the review process.

We look forward to receiving your revised manuscript.

Kind regards,

Mickael Essouma, M. D.

Academic Editor

PLOS ONE

Journal Requirements:

Additional Editor Comments:

The authors have fully addressed my comments about the methodology. However, upon checking the results and discussion, there are some issues that need to be addressed before the manuscript can be accepted for publication. Notably, the total of percentages should be 100 for each variable on Table 1. Did you exclude the Simpson paradox given the lack of effect modification of main study results by age, sex, ethnicity and so on? I suggest separating results of moderation analysis from results of main analyses in tables 3-6. Could you simplify the titles of those tables without mentioning the covariates in titles? Regarding the titles of tables 5 and 6, did you wish to say linear rather than logistic regression?

Reviewers' comments:

Reviewer's Responses to Questions

**Comments to the Author**

Reviewer #1: All comments have been addressed

Reviewer #3: All comments have been addressed

Reviewer #4: All comments have been addressed

2. Is the manuscript technically sound, and do the data support the conclusions?

Reviewer #1: Yes

Reviewer #3: Yes

Reviewer #4: Yes

3. Has the statistical analysis been performed appropriately and rigorously?

Reviewer #1: Yes

Reviewer #3: Yes

Reviewer #4: Yes

4. Have the authors made all data underlying the findings in their manuscript fully available?

Reviewer #1: Yes

Reviewer #3: Yes

Reviewer #4: Yes

5. Is the manuscript presented in an intelligible fashion and written in standard English?

Reviewer #1: Yes

Reviewer #3: Yes

Reviewer #4: Yes

Reviewer #1: Dear Author(s),

This revised manuscript presents a high-quality, methodologically sound, and policy-relevant study that makes a clear contribution to the existing literature. The improvements made in response to prior feedback—especially the enhanced theoretical framing, clearer explanation of variables, and expanded discussion—demonstrate great effort and scholarly care. The paper is now well-structured, coherent, and supported by strong empirical evidence. This is a strong, well-developed manuscript. I commend the authors for their thorough revisions and recommend the paper for publication in its current form.

Sincerely,

Reviewer #3: The authors reflect each of all my review comments logically and rigorously. Therefore, I believe this manuscript is publishable to international readers. I hope this manuscript is cited and read by many readers across the globe.

Reviewer #4: Dear Editor,

Thank you for having given to me the opportunity to review the manuscript titled "The Moderating Role of Regional Political Climate on the Association Between Political Conservatism and COVID-19 Vaccine Hesitancy in the United States." This article investigates whether local political climate, measured by Republican vote share at the ZIP code level in the 2020 presidential election, moderates the association between self-reported political conservatism and COVID-19 vaccine hesitancy and uptake. The authors find that progressives exhibit low hesitancy and relatively high vaccine uptake across all contexts, while conservatives are less hesitant and more likely to receive a response in more progressive environments than in conservative ones.

The topic is highly timely and important, and the manuscript has several strengths: the research question is clearly articulated, the analytical strategy is transparent, and the psychometric properties of the hesitancy scales are robust. The provision of open data and materials, often overlooked in survey-based research, is certainly an excellent foundation.

Although the article has improved, thanks in part to previous comments, I believe its overall contribution is modest, and several conceptual and methodological issues substantially limit the conclusions that can be drawn from the results.

First, the central model, according to which conservatism and a conservative regional context are associated with greater COVID-19 vaccine hesitancy, is already well documented in the literature (I don't think it's necessary to cite previous studies published on the topic here, given their vast and therefore easily accessible nature). The manuscript's claim to novelty likely rests on two aspects. (a) The interaction between ideology and local political climate. These interaction effects are statistically significant but small, derived from self-reported cross-sectional data in a non-probability sample and estimated with a crude ecological proxy for the context. The article is not fully convinced that this adds more than an incremental refinement to what is already known. (b) Certainly more interesting, and the main contribution to the article, is to have identified how those politically aligned to the right, or at least with conservative views, tend to be less hesitant, and therefore more willing, to vaccinate if they live in more progressive areas. This is likely related to social pressure from the context, which would explain this result, which is certainly relevant to note.

Second, the operationalization of "political climate" as a Republican vote share is understandable but rather limited. This variable almost certainly combines political norms with structural and demographic characteristics such as urbanity, racial composition, socioeconomic status, and access to healthcare. No ecological controls are included in the control variables, and the analysis does not account for clustering by geographic area. This raises serious concerns about ecological confounding and underestimated standard errors, weakening any interpretation of climate as an independent contextual influence on individuals.

Third, the sampling strategy and the resulting analytic sample limit generalizability. The study uses a non-probabilistic online panel, excludes a large fraction of initial respondents, and then further restricts the sample to Black and White participants with valid zip codes. These choices are not unreasonable, but they require a much more explicit discussion of the selection processes and the limitations of demographic inference than currently provided.

Finally, some aspects of the structure and interpretation require revision. In several places, the manuscript lapses into causal and perhaps politically biased language that is not supported by cross-sectional observational data, as it should be. The suggested implications for the targeting of conservatives in progressive regions are potential and should be clearly formulated as hypotheses for future investigations rather than as certainties requiring further research. The authors rightly emphasize that the data are cross-sectional and that causal inferences cannot be drawn; they explicitly state that “we cannot assume causal or directional associations.” However, causal language still appears intermittently, for example in phrases such as “the impact of liberal political climates on conservative individuals has significant implications” in the abstract. I would recommend revising the manuscript to replace such language with more neutral formulations (“association,” “pattern,” “relationship”) and to avoid implying that climates influence or exert effects on individuals in the absence of longitudinal or quasi-experimental evidence.

This is not to suggest that this is a fatal flaw that absolutely precludes publication. The article is well-argued, follows academic standards, and offers new, interesting, and partly original positions and arguments, as an article should. As mentioned, I would recommend a light revision, with particular attention to clarifying the modest incremental contribution and broadening and deepening the discussion of limitations.

**Do you want your identity to be public for this peer review?** For information about this choice, including consent withdrawal, please see our Privacy Policy

Reviewer #1: No

Reviewer #3: No

Reviewer #4: **Yes:** Frans Lavdari

---

## [Author Response · Author response to Decision Letter 2]

8 Jan 2026

All reviewer comments have been addressed in the attached Response to Review file.

---

## [Editor Report · Decision Letter 2]

18 Jan 2026

Regional political climate's moderating role in the association between political conservatism and COVID-19 vaccine hesitancy in the United States

PONE-D-25-30422R2

Dear Dr. Kmush,

We’re pleased to inform you that your manuscript has been judged scientifically suitable for publication and will be formally accepted for publication once it meets all outstanding technical requirements. We thank you for continuing to improve upon your article so that we can publish high-quality content in PLOS One.

Kind regards,

Mickael Essouma, M. D.

Academic Editor

PLOS One

Additional Editor Comments (optional):

there is a typo in line 120.

In tables 3 and 4, consider replacing B with β.

In tables 5 and 6 and in the title of table 6, «vaccine hesitancy» should be replaced with «vaccine status», and «booster hesitancy» should be replaced with «booster status».
---

## [Editor Report · Acceptance letter]

PONE-D-25-30422R2

PLOS One

Dear Dr. Kmush,

I'm pleased to inform you that your manuscript has been deemed suitable for publication in PLOS One. Congratulations! Your manuscript is now being handed over to our production team.

Kind regards,

on behalf of

Dr. Mickael Essouma

Academic Editor

PLOS One